# Negative Pressure Wound Therapy in Maxillofacial Applications

**DOI:** 10.3390/dj4030030

**Published:** 2016-09-06

**Authors:** Adam J. Mellott, David S. Zamierowski, Brian T. Andrews

**Affiliations:** Department of Plastic Surgery, University of Kansas Medical Center, Kansas City, KS 66160, USA; amellott@kumc.edu (A.J.M.); davezam1@aol.com (D.S.Z.)

**Keywords:** negative pressure wound therapy, vacuum assisted closure, maxillofacial, craniofacial, wound healing

## Abstract

Negative pressure wound therapy has greatly advanced the field of wound healing for nearly two decades, by providing a robust surgical adjunct technique for accelerating wound closure in acute and chronic wounds. However, the application of negative pressure wound therapy in maxillofacial applications has been relatively under utilized as a result of the physical articulations and contours of the head and neck that make it challenging to obtain an airtight seal for different negative pressure wound therapy systems. Adapting negative pressure wound therapies for maxillofacial applications could yield significant enhancement of wound closure in maxillofacial applications. The current review summarizes the basic science underlying negative pressure wound therapy, as well as specific maxillofacial procedures that could benefit from negative pressure wound therapy.

## 1. Introduction

Negative pressure wound therapy (NPWT) is the application of a continuous or intermittent subatmospheric pressure to a localized wound environment using a topical negative pressure dressing (TNPD) connected to a vacuum pump [1,2]. TNPD are typically open-cell reticulated foams, polyurethane or other material, or gauze-based vacuum dressing. NPWT is also known as subatmospheric pressure (SAP), topical negative pressure (TNP), vacuum-assisted closure (V.A.C.), and microdeformational wound therapy (MDWT), and has greatly impacted the field of wound and surgical care over the past nearly 20 years [3]. NPWT has been used extensively in the treatment of acute and chronic wounds on the torso and limbs, and has been used to treat diabetic foot ulcers [4,5,6,7]. However, NPWT has been under utilized in head and neck surgeries, despite several case studies showing safe and effective use of NPWT [8,9,10]. The current manuscript will briefly review the history of wound healing, the basic science behind NPWT, the considerations for maxillofacial applications, and future improvements for NPWT.

### 1.1. Wound Healing

To first understand how NPWT augments wound healing, it is helpful to briefly review the primary phases of wound healing, which have been review extensively elsewhere [11,12,13,14]. After the body sustains injury via trauma or surgical manipulation, the wound healing response (Figure 1) initiates [15]. In minor wounds the first phase of wound healing starts with hemostasis where platelets begin to accumulate at the injury site, and begin to stick together to form a clot via the release of fibrin [16]. The clot serves to temporarily plug the wound site to slow or prevent further bleeding. Additionally, platelets release cytokines and growth factors, such as platelet-derived growth factor (PDGF), which stimulates recruitment of neutrophils, macrophages, fibroblasts, and myofibroblasts via chemotaxis [17]. The second phase of wound healing, inflammation, starts with the aggregation of neutrophils at the site of injury within the first 24 h of injury [18]. The neutrophils work to eliminate foreign material, bacteria, injured cells, and damaged matrix components within the wound site in addition to mast cells and monocytes, while macrophages are recruited to the wound site by T-cells [19,20]. The activation of macrophages leads to the release of PDGF and transforming growth factor-beta (TGF-β) [21]. The expression of TGF-β is critical in leading to the initiation of the third phase, proliferation. In the proliferation phase, collagen is deposited and cross-linked by enzymes, such as lysyl oxidase [22]. Additionally, angiogenesis initiates enabling the formation of granulation tissue [23,24]. The epidermis (or the intraoral mucosa) begins to re-epithelialize. The fibroblast is primarily responsible for producing new extracellular matrix components such as collagens and glycosaminoglycans (GAGs) to stabilize the structure and mechanical integrity of the tissue. In the final phase, remodeling, wound contracture and scar formation initiate. Excess cells undergo apoptosis, while collagen fibrils, according to Clark et al. [25] orient with the initial alignment of fibrin fibers, which may be influenced by mechanical tension acting directly on the wound site [26]. Myofibroblasts synthesize alpha-smooth muscle actin (α-SMA), which enables collagen to contract and pull wound edges together [27]. Cellular debris and compromised extracellular matrix fibers are destroyed by matrix metalloproteinase (MMPs) produced by fibroblasts, endothelial cells, and macrophages [28]. Disruptions in remodeling where excess collagen is deposited can lead to hypertrophic or keloid scar formation [29,30,31].

### 1.2. Wound Dressings

Since ancient times, wound dressings have been used, and have traditionally been made out of cotton and more recently nylon gauze. Gauze dressings are hydrophilic, used as a wound contact layer, and to remove exudate, and as a barrier to outside contamination. In 1962, Dr. George D. Winter demonstrated in his seminal study that the use of a polyethylene film as a wound dressing, helped keep the wound site moist, which reduced the time for re-epithelialization [32]. Dr. Winter’s study emphasized the importance of keeping the wound site moist and spurred the development and investigation of new materials, including custom polymers (e.g., polyvinyl alcohol (PVA), polyethylene glycol (PEG), polyurethane (PUR), etc.), biological materials (e.g., collagens, carboxymethyl cellulose, alginate, hyaluronic acid, etc.), and composite materials in the forms of woven-meshes and foams of any imaginable pore size and shape, films, and hydrogels [33,34,35,36]. In addition, as the development and construction of biomaterials has advanced for medical applications, dressings have been infused with different antimicrobial agents, such as silver or iodine, and have been designed to work in concert with topical therapies, negative pressure therapies, or instillation [37]. Li et al. [38] reported that silver nanoparticles can be used to destroy the bacterial walls and prevent bacterial replication, and do not exhibit any toxic effects on mammalian cells. Furthermore, technologies, such as nanofiber electrospinning and precision particle fabrication, enable the ability to precisely load different polymers with antimicrobial agents, growth factors, or vitamins to actively interact with the wound environment [39,40,41]. Currently, construction and selection of dressings focus on maintaining a moist wound environment, decreasing bacterial burden, and stimulating cell proliferation to aide the wound healing progression [42,43,44].

### 1.3. Vacuum Assisted Closure

In 1997, building on advancements in wound dressings and drains, Drs. Argenta and Morykwas introduced a groundbreaking device that applied a subatmospheric pressure through an open-cell reticulated foam dressing to acute, sub-acute, and chronic wounds in both an animal and clinical study [45,46]. The subatmospheric device used by Drs. Argenta and Morykwas dramatically accelerated formation of granulation tissue. The dressing system kept wound environments moist while reducing edema/exudate, promoting angiogenesis, and possibly reducing bacterial burden. The dressing system was further developed, refined, and termed “Vacuum Assisted Closure (V.A.C.)” by Kinetic Concepts Inc. (KCI, San Antonio, TX, USA), and distributed as the first NPWT by KCI. The V.A.C. system has had a tremendous impact on the field of wound healing [47,48,49]. The V.A.C. system is comprised of five basic components: (1) The open reticulated polyurethane ether foam dressing (GranuFoam™); (2) semi-permeable adhesive film; (3) suction tubing; (4) collection canister; and (5) vacuum pump (Figure 2). The GranuFoam™ is traditionally custom cut to fit the wound site by the healthcare provider and placed within the wound so that GranuFoam™ completely interfaces with the wound bed. The semi-permeable occlusive film drape is adhesively secured around and over the wound site and the GranuFoam™ creating an airtight seal. A small opening is made in the semi-permeable film drape, and the suction tubing and its port are placed at the opening. The suction tubing is attached to the collection canister, which is fitted to the vacuum pump. When the vacuum pump is activated, a suction differential of 75 to 125 mmHg is applied to the entire wound surface. Under the vacuum, the GranuFoam™ compresses, which results in wound edges being pulled closer together macroscopically while also causing microdeformations at the interface between the GranuFoam™ and wound tissue, which stimulates the healing process [50]. The mechanical forces, wound-dressing selection, and microenvironment interactions all contribute the success of V.A.C., and the primary mechanisms of action will be further expounded on to provide a context for increasing the use of V.A.C. in maxillofacial applications.

## 2. V.A.C. Mechanisms of Action

The core events that occur during NPWT and affect the process of wound healing include: (1) macrodeformation or shrinkage of the wound; (2) microdeformation at the wound-dressing interface exerting strain on the extracellular components of the wound site; (3) exudate fluid removal and edema reduction; and (4) possible reduction of infectious material (Figure 3). Additionally, there are secondary events that occur that aid in stabilizing the wound environment and accelerate the wound healing process, such as mechanotransduction, moisture control, temperature stabilization, cell recruitment, and cell proliferation. Together, the aforementioned events affect the inflammation, proliferation, and remodeling of the wound site as healing occurs.

### 2.1. Macrostrain

Macrostrain or macrodeformation occurs when the TNPD shrinks in size, and the wound edges pull together. The GranuFoam™ is an interconnected “mesh net”, containing open pores ranging in size from 400 to 600 μm in diameter [51]. Scherer et al. [52] demonstrated that GranuFoam™ can shrink by up to 80% of its original volume. The deformation of the TNPD is directly related to the volume of the material that is occupied by air. The pump evacuates the air, which enables shrinkage of the TNPD. Thus, choosing an appropriate TNPD to interface with the wound bed is important for considerations of macrodeformation. However, not all tissues possess the same elasticity properties, which directly limits the degree to which wounds can contract. For example, a wound located on the head, such as the scalp, where the skin is already fixed to fascia, may not contract to the degree that a wound on the abdomen would contract as result of the different tensile and elastic forces acting on the skin in both locations [51,53].

In addition, while a V.A.C. system applies subatmospheric pressure to a wound site, the strain exerted at the wound surface itself may increases pressure in the underlying tissues proportional to the suction force. Kairinos et al. [54,55,56] reported the detection of increased atmospheric pressure in surrounding tissues in circumferential negative pressure wound therapy dressings; however, Kairinos et al. [54,55,56] noted that the pressure gradually decreased over 48 hours in most wounds analyzed. Hence, caution is advised in applying a V.A.C. system to a wound near or directly over delicate vasculature or internal organs, such as the dura/brain.

### 2.2. Microstrain

Microstrain is the force exerted directly on the cells via the extracellular matrix from the shrinkage of the TNPD in the wound. The vacuum creates a microdeformation in which cells are pulled into the pores of the foam while the struts provide an equal and opposing force that pushes cells away from the foam, creating a quilting pattern at the wound interface [57]. Microdeformation is advantageous for the proliferation phase of wound healing as the TNPD microdeformation acts on the extracellular matrix initiating mechanosignaling within the fibroblasts, epithelial cells, and myofibroblasts to stimulate cell proliferation, differentiation, angiogenesis, and neurogenesis [58]. Saxena and colleagues reported a measured tissue strain on the surface of the wound of 5%–20% using finite element analysis [59].

### 2.3. Fluid Removal and Edema Reduction

Edema is an excess interstitial fluid trapped within tissue, which manifests as swelling. Edema can be deleterious to the wound healing process, as the extracellular fluids can exert their own compressive forces on microvasculature and cells. This can stall cell proliferation and reduce blood flow, as well as increasing the diffusion distance for nutrients [60,61]. When the subatmospheric pressure is applied to a porous TNPD, the excess fluid can be evacuated out. The reduction of edema and compressive burden combined with the stimulation of mechanical signaling enables cells to proliferate [62]. Additionally, the evacuation of fluid may reduce the toxins, cell debris, and disrupted bacteria from the wound site, while increasing the efficiency of the volume of fluid drained by the lymphatic system by increasing blood perfusion.

### 2.4. Reduction of Infectious Material

The reduction of infectious material within the wound site as a result of the application of V.A.C. is controversial [63]. During the inflammation phase of wound healing, neutrophils, macrophages, and lymphocytes work together to destroy foreign agents in the wound site to prevent excessive invasion of pathogenic bacteria. However, if the bacterial burden is too high, and pathogenic bacteria species invade the epithelium and disrupt the expression of MMPs and cytokines that promote inflammation, effectively creating a continuous inflammation cycle [64]. The wound microenvironment is dynamic and factors, such as moisture, pH, oxidation, and surface area, play a critical role in the survival and invasion of pathogenic bacteria. Gram-positive bacteria, such as *Staphylococcus aureus*, grow at an optimal pH of 6–7, while *Pseudomonas aeruginosa* and *Enterococcus faecalis* are capable of growing at wider pH levels [65]. Wounds that contain foreign debris, eschar, or necrotic tissues are at higher risk of bacterial infection as wound contaminates can provide attachment sites, nutrition, or both for foreign bacteria [66]. Thus, cleaning the wound is important for limiting bacterial infection, and aiding wound healing. Mertz et al. [67] reported a reduction in endogenous Gram-positive bacteria, but not Gram-negative bacteria after 5 days when a semi-occlusive polyurethane film dressing was used in partial thickness excisional wounds created on the back of pigs. In the original pig model study published by Morykwas et al. [46] wound defects were inoculated with 10^8^ organisms of a human isolate of *Staphylococcus aureus* and a pig isolate of *Staphylococcus epidermis*. After 5 days of NPWT, bacterial counts were reduced to fewer than 10^5^ organisms per gram of tissue. However, Mouës et al. [68] reported no significant differences in bacterial counts between patients that received NPWT and patients that received standard therapy where a moist gauze dressing was changed twice per day. Mouës et al. [68] have proposed that difference in bacterial counts may be a result of using a biopsy sampling technique instead of superficial swabbing as was used in previous studies. The use of GranuFoam™ impregnated with silver nanoparticles (GranuFoam Silver™) has been reported to reduce bacterial load in chronic wounds [69]. Further examination of bacterial behavior and bacterial quantification within wound sites is still required to fully understand if or how the NPWT is directly reducing bacterial load.

### 2.5. Wound Stabilization and Secondary Events

The removal of fluid and reduction of edema can act as a micro-debridement of the wound tissue, which can be further enhanced by actual irrigation through a V.A.C. system. This allows cells to re-establish homeostatic osmotic and oncotic gradients, in addition to establishing hypoxic conditions that produce vascular endothelial growth factor (VEGF) gradients that direct angiogenesis [70]. Increased blood flow and perfusion allow for the critical supply of nutrients to stimulate cell proliferation and extracellular matrix remodeling [71]. Furthermore, the semi-permeable occlusive drape isolates the wound site, and prevents contamination as well as acting as a thermal insulator to keep wound temperatures optimal for healing. The semi-permeable drape in combination with a hydrophobic foam core additionally prevents evaporative water loss, which is critical for keeping the wound moist for cell migration and transport of nutrients. The combination of macrostrain and microstrain act on cell shape, and induce mechanotransduction that can affect cell behavior and expression of critical signals that promote cell proliferation, extracellular matrix deposition, and remodeling [59]. Nuutila et al. [72], were the first group to report increases in gene expression of inflammation signals (Interleukin 8 (IL8), IL24) and tissue remodeling signals (MMP1, MMP3, and MMP10) in a clinical study in which patients were treated with NPWT. In addition, to wound stabilization some studies have reported a reduction in the wound closure duration. Arti et al. [73] reported a 1.5 day reduction in the duration of hospital stay and 19% reduction in wound surface for patients receiving NPWT for skin graft or flap coverage as opposed to patients receiving conventional wound dressings. In a separate study, a 13.7 day reduction in the duration of wound closure for cats with open wounds was reported [74]. Additional studies examining treatment of open-wounds with NPWT against other conventional therapies could be beneficial in determining the effect of NPWT on wound closure duration.

## 3. Maxillofacial Considerations

The head and neck provides many unique problems to maxillofacial surgeons in the setting of wound healing and reconstruction. Function and aesthetics are often equally important to most patients. Simultaneous management of both function and aesthetics has driven a large market for products such as TNPDs used in NPWT.

Use of NPWT as a surgical adjuvant in maxillofacial surgery was first described in 2006 [75,76]. This paper retrospectively reviewed the use of NPWT in several complicated maxillofacial situations, such as exposed calvarial bone, bolster dressing for large facial skin grafts, and wound management following necrotizing fasciitis debridement. NPWT was successful in all such clinical situations and this work led to other published reports that expanded its use.

Palm et al. [77] reviewed 1502 peer-reviewed journal articles on “vacuum therapy” for which 37 articles pertained to maxillofacial surgery. They noted that studies were generally limited by containing case reports or case series and NPWT was used as an adjuvant maxillofacial reconstructive procedures and management of soft tissue defects of the neck. However, published reports on NPWT use are in all areas of maxillofacial reconstruction. The largest study by Satteson et al. [78] reviewed 69 patients with 73 head and neck wounds resulting from cancer (86%), trauma (8%), infection (3%), or burns (3%) that used V.A.C. in conjunction with skin grafts, Integra, and open debrided wounds. Minor complications were reported in 56% of patients that received skin grafts, 33% of patients that received Integra, and 29% of patients with open debrided wounds. Most complications were resolved with follow-up treatment.

### 3.1. Upper Third Maxillofacial Reconstruction

NPWT use for scalp and forehead reconstruction was first described by Andrews et al. [75]. Its use for skin and soft tissue defects has been further described by Hsia et al. [79]. Both studies demonstrated that NPWT could be used to temporize wounds in the setting of trauma and contamination or definitely treat wounds by promoting granulation tissue formation and wound contracture/epithelialization. More aggressive protocols have been used to manage exposed dura and/or brain when the calvarium is missing [80,81,82]. Ahmed et al. [80] discussed technical nuances to manage cerebrospinal fluid (CSF) compartmentalization and eventual soft tissue coverage of dural repairs in their case report.

### 3.2. Middle Third Maxillofacial Reconstruction

Soft tissue defects of the cheeks and orbit are particularly suitable for NPWT. NPWT can be used to promote granulation tissue formation on the facial skeleton so that skin grafts can be applied, for wound management and contracture, or for skin graft bolstering [75]. Although these applications seem straightforward concerns arise in the midface maintaining an occlusive seal of any negative pressure device secondary to the midface contours and the presence of the eyes, nose, mouth and ears.

### 3.3. Lower Third Maxillofacial Reconstruction

NPWT use on the lower jaw and its soft tissue has been well described. Zhang et al. [83] described its use in the management of submandibular fistulas after osteoradionecrosis reconstruction. In their study NPWT was successful closing small submandibular fistulas that developed in nine patients in 7 to 12 days. The proximity of the oral cavity and often a tracheostomy make utilization of NPWT in this region difficult. 

### 3.4. Neck Reconstruction

Cutaneous-oral fistulas are a rare and difficult complication of maxillofacial surgery. TNPDs have been shown to be an effective means of closing these intraoral communications [75]. Long track fistulas with collapsible non-radiated tissues have been shown to be most amenable to this mode of closure. Yang et al. [84] used TNPDs to successfully close eight salivary fistulas with an average time of treatment being 10.8 days. Tian et al. [85] further advanced the utilization of TNPD for salivary fistula closure by employing dental paste intraorally to maintain an occlusive seal for the negative pressure system. In their series, they successfully close salivary fistulas in 9 of 10 patients.

### 3.5. Technical Considerations

Establishing an occlusive environment and maintaining a negative pressure seal is one significant obstacle in NPWT head and neck utilization. Occlusive dressings that avoid important functional structures such as the eyes, ears, nose, and mouth should be employed. If necessary these structures can be temporarily covered; however, they should be protected and direct negative pressure application minimized to these structures (especially the eyes). Although NPWT has been used on exposed brain/dura caution should be used in this application. A watertight CSF closure should be ensured prior to its use as to not immediately empty the CSF reserves and cause herniation. Frequent dressing changes in NPWT are needed in the head and neck and are often done daily at our institution to maintain the occlusive seal. Currently, TNPDs for NPWT are manufactured as flat elliptical foam sponges, most conducive for treating wounds located on the abdomen or limbs. Interestingly, healthcare providers have successfully used NPWT and TNPDs to treat diabetic foot ulcers, which may provide insights for using NPWT on wounds occurring on the head and neck. The ability to make a more malleable or shapeable foam to fit the contours of the head and neck, will allow significant opportunities to more easily utilize NPWT in maxillofacial applications.

## 4. Conclusions

NPWT accelerates wound healing by enhancing the inflammation, proliferation, and remodeling phases of wound healing. The pressure differential facilitates movement of fluid and reduces edema while increasing blood flow, which aids in lymphatic drainage and clearing of the wound to reduce inflammation. Simultaneously, the macrostrain and microstrain initiate mechanotransduction, which stimulates fibroblasts, myofibroblasts, and epithelia cells to proliferate while building new extracellular matrix and releasing enzymes to remodel and process the developing support matrix. In conjunction, VEGF gradients are established as a result of hypoxia, and direct angiogenesis, which further increases blood perfusion, and contributes to repairing the original tissue. Healthcare providers have a well-established history of treating acute and chronic wounds occurring on the torso and limbs with NPWT. However, while NPWT has been shown to be safely and effectively used in complicated maxillofacial wounds, NPWT is under utilized in this due to the limitations in maintaining a negative pressure seal when TNPDs are applied to the head and neck. Further work remains to develop next generation TNPDs to fit the contours of the head and neck. These technologies have the potential to increase the use of NPWT in maxillofacial applications, thereby benefitting patients with a wider range of injuries.

## Figures and Tables

**Figure 1 dentistry-04-00030-f001:**
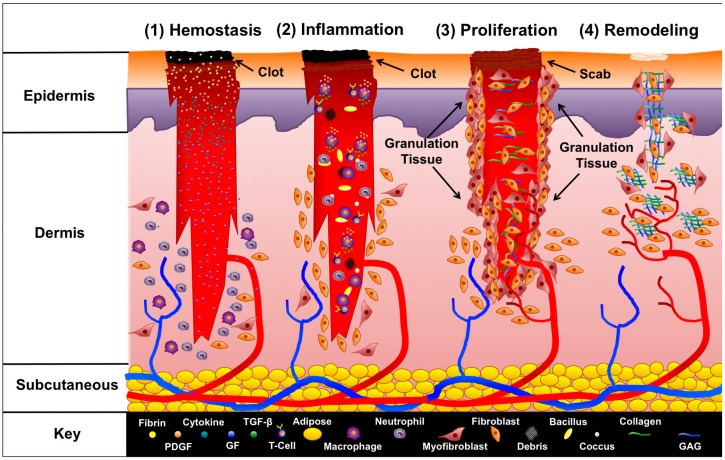
**Phases of wound healing.** The four phases of wound healing are illustrated. Immediately after a wound forms, hemostasis begins. Platelets form a clot, and are bound together by fibrin. Several cytokines are released, which recruit neutrophils and other leukocytes to the site of injury to start the inflammation phase. Leukocytes begin clearing the wound of bacteria, debris, and other foreign contaminants. T-cells infiltrate the wound and recruit macrophages, which release PDGF and TGF-β to signal fibroblasts and myofibroblasts to start the proliferation phase. Granulation tissue forms, and fibroblasts begin developing new extracellular fibers by producing collagen and GAGs. In addition angiogenesis begins, and new blood is supplied to the site of injury. After proliferation, the final phase, remodeling, occurs in which extracellular fibers align, the wound contracts, and fibroblasts release enzymes to remove damaged extraneous extracellular matrix. Epithelial cells, despite their defining role in coverage, healing, and controlling “crosstalk” with fibroblasts, have been deliberately left out of this schematic representation for the sake of clarity. (PDGF = Platelet-Derived Growth Factor, GF = Growth Factor, GAG = Glycosaminoglycan).

**Figure 2 dentistry-04-00030-f002:**
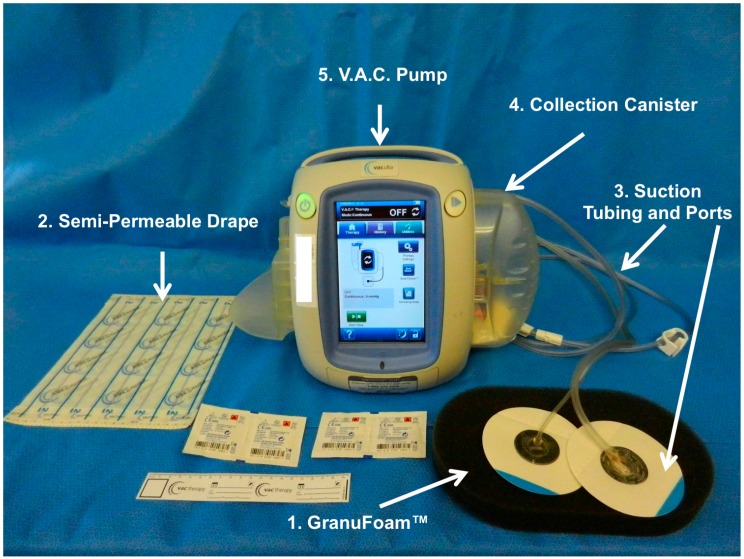
**Vacuum Assisted Closure (V.A.C.) system.** KCI Veraflow™ is the flagship model, and is designed to provide instillation therapy in addition to standard V.A.C. therapy. The (1) black semi-occlusive dressing (GranuFoam™) along with the (2) semi-permeable drape used to isolate the wound and prevent escape of evaporative moisture. In addition, the (3) suction tubing and suction ports are connected to the (4) collection canister through which (5) the vacuum pump exerts suction force.

**Figure 3 dentistry-04-00030-f003:**
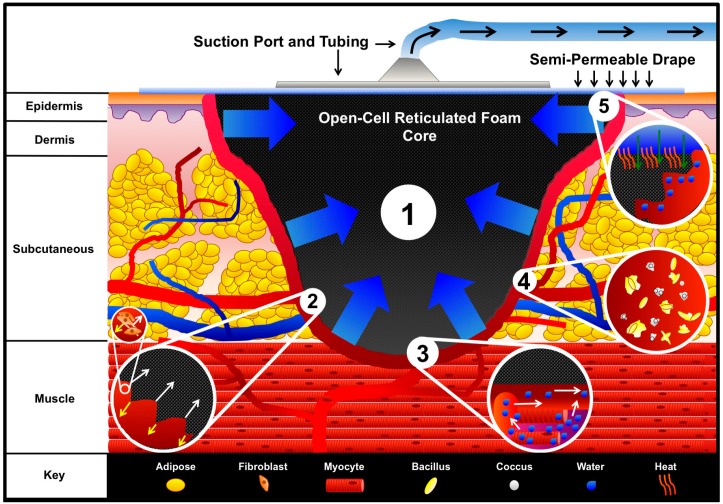
**V.A.C. mechanisms of action.** An open-cell reticulated foam is placed within the wound and covered by a semi-permeable adhesive drape. The drape is adhesively secured around and over the wound to create an airtight seal. A small hole is made within the center of the drape, and the suction port and tubing connected to the collection canister are attached. Engaging the vacuum pump evacuates the air from the foam and enables (1) Macrodeformation of the foam via shrinkage, which pulls the wound edges together. At the interface between the foam and wound bed, (2) microstrain occurs in which cells are pulled into the pores of the foam while an equal and opposing force acting on the struts of the foam pushes cells away. The microstrain on the cells initiates mechanotransduction, which can stimulate cell proliferation as illustrated within the sub-inset of inset 2. Additionally, engagement of the vacuum facilitates the (3) movement of fluid out of interstitial spaces, thereby reducing edema and increasing blood flow as illustrated in inset 3. The V.A.C. system possibly (4) reduces bacterial burden; however, the mechanism by which bacteria are reduced is not fully understood. The destruction of bacteria is illustrated in inset 4. The V.A.C. system contributes to (5) wound stabilization through secondary events. Inset 5 illustrates the movement of warm air down through the semi-permeable drape into the wound space, while isolating the wound from foreign contaminants. Furthermore, the semi-permeable drape prevents evaporative water loss, which aids in keeping the wound moist to enable cell migration and nutrient transport.

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
