# Peer review of "Negative Pressure Wound Therapy in Maxillofacial Applications"

_dentistry, 2016, doi:10.3390/dj4030030_

Round 1

Reviewer 1 Report

needs figures from an open access journal or authors

Argenta did not invent this, in use for decades in makeshifg abd wound dressings, he just patented it and made money

Simplify and reduce mechanisms discussed, except wound contraction and granulation tissue nothing is really certain about how it works.

Not all references are correct

Reviewer 2 Report

This is a well written and comprehensive review on a topic interesting to the reader of the Dentistry Journal.  Below are my comments:

L. 56: “collagen fibrils align according to mechanical tension”.  RAF Clark has proposed that collagen fibrils align according to the initial alignment of fibrin fibers [Clark RAF (1985) Cutaneous tissue repair: Basic biologic considerations. I. J. Am. Acad. Dermatol. 13: 701-725].  This possibility should be mentioned.

L. 55: "In the final phase, remodeling, digestion and crosslinking of collagen fibrils occur via collagenase enzymes [22]".  First, I am not aware that remodeling of collagen involves action of collagenases.  Second, cross-linking involves cross-linking enzymes that typically act in concert with production of collagen (not at later times).  Third, reference 22 mentions neither collagenase nor cross-linking.  Please revise statement and citation.

L. 45 etc.: description of the inflammatory cell types is currently a little “old school”. An important role for T cell in mediating macrophage recruitment etc. is now broadly recognized and should be mentioned.  T cells are also missing from Fig.1.

Fig.1: what is the rationale/evidence for representing myofibroblasts at the periphery of the granulation tissue (as opposed to throughout)?

There is no indication of the potential benefit of using NPWT on wound closure duration.  Please comment on this and describe what the expectations should be (reduction by x%?)

L. 214 and ref.73: detail the study a little more (e.g. this studies shows great discrepancy between VAC applications —56% complications for grafts vs 29% complications when used on debrided wounds.  These differences might be clinically important).

Round 2

Reviewer 2 Report

The authors have responded to my comments in a satisfactory way.  I have no further comments.